# Development of a Microwave Irradiation Probe for a Cylindrical Applicator

**Tomohiko Mitani [1],\*** 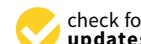**, Ryo Nakajima [1], Naoki Shinohara [1], Yoshihiro Nozaki [2], Tsukasa Chikata [2] and Takashi Watanabe [1]**

[1] Research Institute for Sustainable Humanosphere, Kyoto University, Kyoto 611-0011, Japan; ryo_nakajima@rish.kyoto-u.ac.jp (R.N.); shino@rish.kyoto-u.ac.jp (N.S.); twatanab@rish.kyoto-u.ac.jp (T.W.)

[2] Japan Chemical Engineering & Machinery Co. Ltd., Osaka 532-0031, Japan; y-nozaki@nikkaki.co.jp (Y.N.); chikata9366@gmail.com (T.C.)

\* Correspondence: mitani@rish.kyoto-u.ac.jp; Tel.: +81-774-38-3880

**Abstract:** A microwave irradiation probe was newly developed for downsizing microwave applicators and the overall microwave heating apparatus. The key component of the proposed probe is a tapered section composed of polytetrafluoroethylene (PTFE) and alumina. Insertion of the tapered section between the input port and the applicator vessel realizes impedance matching to the microwave power source and reduces the reflected power from the applicator. The proposed microwave probe for a cylindrical applicator was designed using 3D electromagnetic simulations. The permittivity data of two liquid samples—ultrapure water and 2 M NaOH solution—were measured and taken into simulations. The conductivity of the NaOH solution was estimated from the measurement results. The measured reflection ratio of the fabricated applicator was in good accordance with the simulated one. The frequency ranges in which the measured reflection ratio was less than 10% were from 1.45 GHz to 2.7 GHz when using water and from 1.6 GHz to 2.7 GHz when using the NaOH solution as the sample. The heating rate of the applicator was roughly estimated as 63 to 69 K for a 5 min interval during the 2.45 GHz microwave irradiation at the input power of 100 W.

**Keywords:** microwave heating; applicator design; electromagnetic simulation; coaxial feeding

## 1. Introduction

Applications of microwave heating and microwave irradiation to chemical reactions covering a wide variety of research fields have been reported over the past three decades [1–9]. Various concepts of microwave irradiation methods have been introduced [3], and in most cases, a single-mode cavity and a multi-mode cavity, such as a microwave oven, have been used for the microwave heating applications. Recently, continuous-flow systems have been well studied [1] because the penetration depth of the microwave at 2.45 GHz, which is allocated in the industrial, scientific, and medical (ISM) bands, limits the applicator size of microwave heating. Downsizing the applicator will, therefore, prevail in future microwave heating applications. A number of types of small diameter micro-flow applicators have already been reported [10–13].

Another trend in microwave heating applications is the replacement of the microwave power source. Since the early days of microwave heating, a vacuum tube device called a magnetron [14] has been used as the microwave power source and has also been widely used in microwave ovens. However, high-power solid-state devices are attracting a great deal of attention as an alternative microwave power source due to their variability in output power, adjustability in frequency, and easy operability. Moreover, the solid-state generators can set the frequency precisely and be adjustable with high purity and a narrow bandwidth of frequency spectrum. Although the magnetron is still superior to the solid-state device with respect to cost and maximum output power, the development of solid-state microwave power amplifiers, 1 kW in the 2.45-GHz band [15] and 64 kW in the 915-MHz band [16], were recently commercialized. A number of studies on microwave heating using solid-state devices have also been reported [17–20]. Our group has previously developed a wideband microwave applicator with a coaxial cable structure [21], based on using solid-state devices.

The objective of the present study is to develop a microwave irradiation probe for a cylindrical metal applicator for microwave heating by solid-state devices. In most conventional applicators for microwave chemistry, heated liquid samples were put in a material transparent to microwaves, such as glass [22] or polymer [23]. As a metal resonant cavity and waveguides were used in these types of applicators, the overall microwave heating apparatus became large. The novelty of the proposed system is direct microwave irradiation of a liquid sample in a metal vessel; it also contributes to downsizing the applicator and the overall apparatus compared to the conventional systems. The proposed microwave probe for a cylindrical applicator was designed using 3D electromagnetic simulations based on the finite element method (FEM), which is commonly available and is used in applicator design [24,25].

## 2. Materials and Methods

### 2.1. Overview of the Proposed Microwave Irradiation Probe and Cylindrical Applicator

A cross-sectional schematic of the proposed microwave irradiation probe and the cylindrical applicator is shown in Figure 1. The applicator is a cylindrical vessel made of SUS 316L non-magnetic stainless steel. Microwaves are radiated to the liquid samples from the side of the vessel. A pressure gauge is attached to the top of the applicator to maintain the internal pressure. The microwave irradiation port consists of a commercially available Type-N connector with a characteristic impedance of 50 Ω. In this applicator design, a magnetic stir bar, which can stir liquid samples, can be placed on the bottom of the vessel.

As a feature of the proposed microwave irradiation probe, a tapered section is inserted between the input port and the vessel to realize impedance matching to the microwave power source and to reduce the reflected power from the applicator. This section is composed of polytetrafluoroethylene (PTFE) and alumina. A concave PTFE component is inserted from the microwave input side, and a convex alumina component is inserted from the applicator side. These parts are embedded in the coaxial section, and the inner conductor passes through them. A stainless-steel cylindrical block, called a supporter, is placed at the end of the alumina block to support the inner conductor and the alumina. The supporter also plays a role in preventing the PTFE from varying its length by thermal expansion during microwave heating.

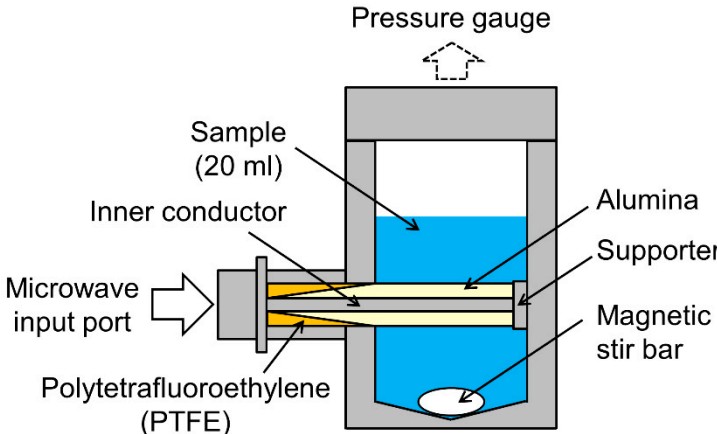

**Figure 1.** Cross-sectional schematic of the proposed microwave irradiation probe and cylindrical applicator. The applicator is made of stainless-steel (SUS 316L). Microwaves are radiated to the liquid samples from the side of the vessel.

## 2.2. Liquid Samples and Permittivity Measurements

Two liquid samples were selected for designing the applicator: ultrapure water (hereinafter water) as a dielectric sample and 2 M NaOH solution (hereinafter NaOH solution) as a dielectric and conductive sample. The water was obtained from an ultrapure water system (PURELAB Flex3 PF3XXXXM1, ELGA, High Wycombe, UK) and had an electrical resistivity greater than 18 MΩ·cm. The NaOH solution was obtained by dissolving NaOH (JIS Special Grade, Wako Pure Chemical Industries, Ltd., Osaka, Japan) in the same water.

Permittivity measurements of the two liquid samples were conducted using the coaxial probe method [26]. The liquid sample was placed in a glass bottle, and a slim form probe (85070E, Agilent, Santa Rosa, CA, USA) was immersed in the sample. The reflection coefficient at the boundary between the sample and the probe was measured by a network analyzer (N5242A, Agilent, Santa Rosa, CA, USA), and the relative complex permittivity $\varepsilon = \varepsilon' - j\varepsilon''$ was calculated over the frequency range from 500 MHz to 20 GHz, where j, $\varepsilon'$ and $\varepsilon''$ are the imaginary unit, the real and imaginary parts of the relative permittivity, respectively. Permittivity measurements were repeated three times at room temperature.

Note that the imaginary part of the measured relative permittivity $\varepsilon''_m$ includes the conductivity $\sigma$ with this measurement method, as expressed by the following equation:

$$\varepsilon''_m = \varepsilon'' + \sigma/(2\pi f \varepsilon_0), \tag{1}$$

where $\varepsilon_0$ and $f$ are the permittivity in a vacuum and the frequency, respectively. The conductivity of the water can be regarded as 0; however, the conductivity of the NaOH solution cannot be ignored when designing the cylindrical applicator with the microwave irradiation probe by electromagnetic simulations precisely. Therefore, we estimated the conductivity $\sigma$ by fitting the inverse proportional curve obtained in the low frequency range in which the effects of $\varepsilon''$ are negligible compared to $\sigma$. Then, the approximate equation of $\sigma$ in the low frequency range is expressed as follows:

$$\sigma \approx 2\pi f \varepsilon_0 \varepsilon''_m. \tag{2}$$

With respect to the water, the measured permittivity data were compared with the Debye relaxation model, as expressed in the following equation:

$$\varepsilon = \varepsilon_\infty + \frac{\varepsilon_s - \varepsilon_\infty}{1 + j2\pi f \tau}, \tag{3}$$

where $\varepsilon_\infty$, $\varepsilon_s$ and $\tau$ are $\varepsilon'$ at sufficiently high frequency, $\varepsilon'$ at sufficiently low frequency and the Debye relaxation time [27]. The parameters of $\varepsilon_\infty$, $\varepsilon_s$ and $\tau$ were obtained from the reference [28]; $\varepsilon_\infty = 5.2$, $\varepsilon_s = 78.36$, and $\tau = 8.27$ ps at a temperature of 25 °C.

### 2.3. Design of the Proposed Microwave Irradiation Probe and Cylindrical Applicator

The proposed microwave irradiation probe and cylindrical applicator were designed with the aid of the 3D electromagnetic simulation software package (HFSS ver 19.0, ANSYS, Canonsburg, PA, USA), which calculates the reflection coefficient, $S_{11}$, for the input microwave port that is defined as the element of the scattering matrix in a one-port network circuit [29].

Simulation models of the applicator are shown in Figure 2. The general simulation setup involved frequencies ranging from 80 MHz to 2.7 GHz with an input microwave power of 100 W. The target frequency in the simulations was 2.45 GHz. The values for the real part of the relative permittivity $\varepsilon'$ and the dielectric loss tangent $\tan \delta = \varepsilon'' / \varepsilon'$ were 2.08 and 0.001 for the PTFE, and 9.4 and 0.006 for the alumina. The conductivity of the SUS 316L was $1.1 \times 10^6$ S/m. The parameters of the PTFE, alumina, and SUS 316L were obtained from the library data in the simulation software. The volume of the liquid sample, either water or the NaOH solution, set in the applicator was 20 mL. The measured relative permittivity and conductivity of the liquid samples were imported into the simulation software. A magnetic stir bar was not taken into account in the simulations. The simulation parameters are summarized in Table 1. The mesh for the simulations was created by using the function of adaptive auto mesh in the simulation software.

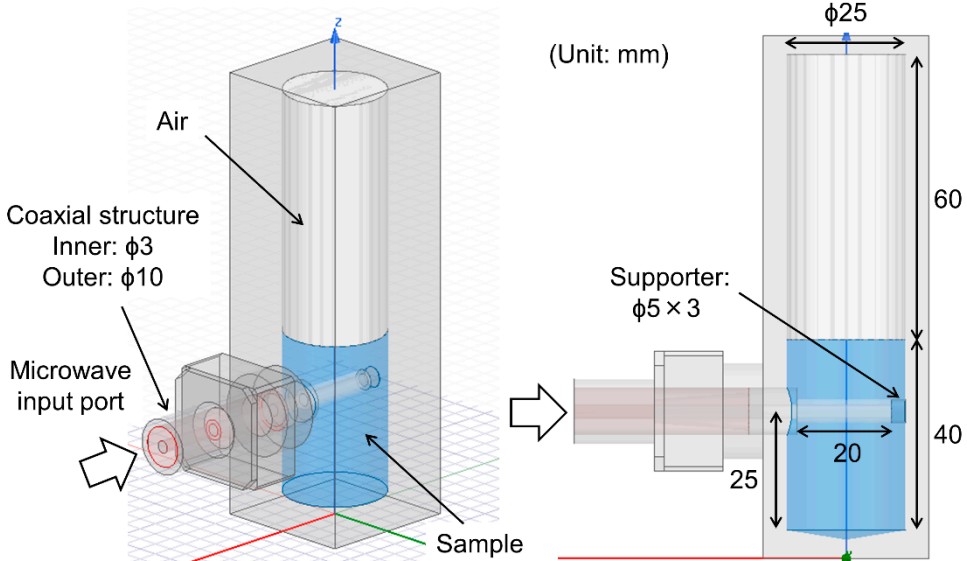

**Figure 2.** Simulation model of the cylindrical applicator with the microwave irradiation probe.

**Table 1.** Simulation parameters.

| Parameters | Values |
| --- | --- |
| Target frequency | 2.45 GHz |
| Start frequency | 80 MHz |
| Stop frequency | 2.7 GHz |
| Input power | 100 W |
| Number of mesh | 19,236 (water), 15,674 (NaOH solution) |
| Permittivity of PTFE | 2.08–j 0.00208 |
| Permittivity of alumina | 9.4–j 0.0564 |
| Conductivity of SUS 316L | $1.1 \times 10^6$ S/m |

Figure 3 shows the designed and fabricated cylindrical applicator with the microwave irradiation probe. The optical fiber thermometer port was attached orthogonally to the microwave input port on the side of the applicator. The pressure gauge can be removed from the applicator to pour in the liquid sample. The maximum temperature and inner pressure were designed to be 200 °C and 2 MPa, respectively.

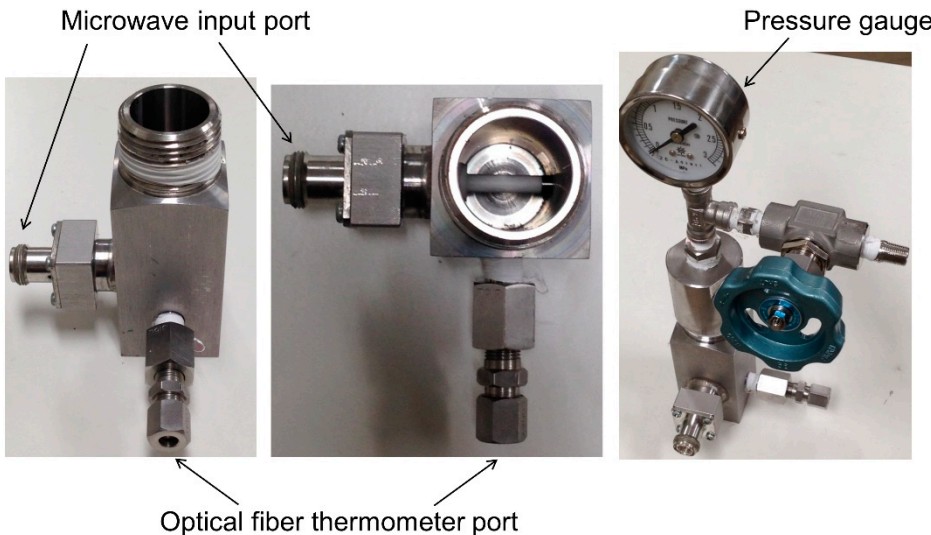

**Figure 3.** Photographs of the developed cylindrical applicator with the microwave irradiation probe. The optical fiber thermometer port was attached orthogonally to the microwave input port on the side of the applicator. The pressure gauge was mounted to the top of the applicator to maintain the internal pressure.

### 2.4. Microwave Reflection Measurements for the Cylindrical Applicator

The reflection coefficient of the microwaves at the input port of the applicator, $S_{11}$, was measured using a network analyzer (Agilent N5242A). The absolute value of $S_{11}$ was converted to the reflection ratio $R$, which is the ratio of the input power to the reflected power, by the following equation:

$$R = 100\% \times |S_{11}|. \tag{4}$$

A 20 mL aliquot of the liquid sample, either the water or the NaOH solution, was poured into the applicator, and microwave reflection measurements were executed at room temperature.

### 2.5. Microwave Heating Tests

A diagram of the microwave heating measurement system is shown in Figure 4. A 20 mL aliquot of the liquid sample was poured into the applicator and stirred during the microwave heating. Microwaves were generated using a signal generator (Agilent N5183A), amplified by a solid-state power amplifier (R&K GA0827-4754-R), and input to the applicator through a coaxial cable. The input power $P_i$ and the reflected power $P_r$ were monitored by a power meter (Agilent E4417A) through power sensors (Agilent N8485A). Here, $P_i$ was fixed at 100 W during the microwave heating tests. The sample temperature $T$ was measured using a fiber optic thermometer (ANRITSU FL-2400). Although a single-point measurement cannot verify the temperature uniformity of the liquid sample even if it is stirred, the measured temperature was used as a representative of sample temperature in this study, due to the limitation of temperature measurements in the applicator. The values of $P_i$, $P_r$, and $T$ were logged at one-second intervals using a data logger (GRAPHTEC GL800). The microwave heating tests were conducted at frequencies of 1.7 GHz and 2.45 GHz. The microwaves were stopped when

the measured temperature reached 100 °C. The reflection ratio $R$ defined by Equation (1) was also obtained as the ratio between $P_i$ and $P_r$ as follows:

$$R = 100\% \times P_r/P_i. \tag{5}$$

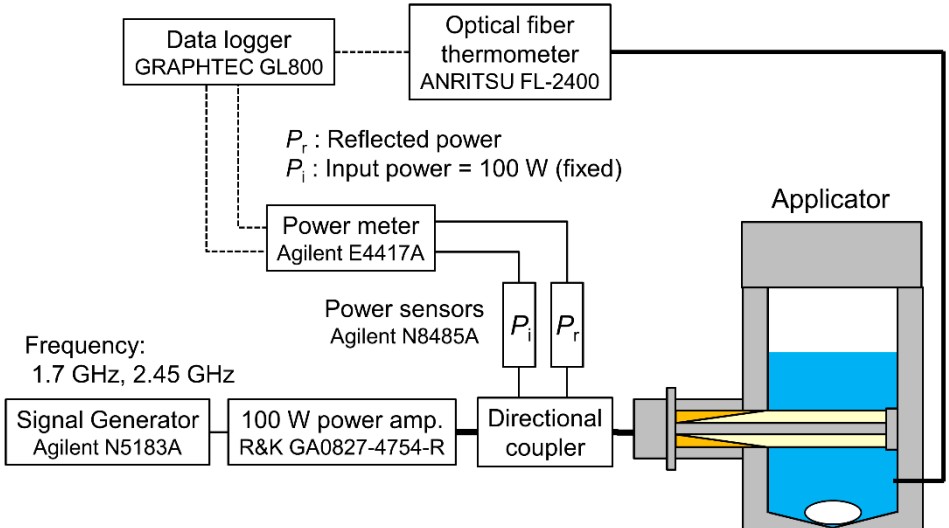

**Figure 4.** Diagram of the microwave heating measurement system.

## 3. Results

Figure 5 shows the measured average permittivity of two liquid samples. When the average permittivity data were regarded as the true permittivity, the measurement errors were less than 1.4% in the water case and less than 1.0% in the NaOH case. Note that the measured data for the NaOH solution in Figure 5b represents the apparent imaginary part of the relative permittivity because they include the conductivity of the solution.

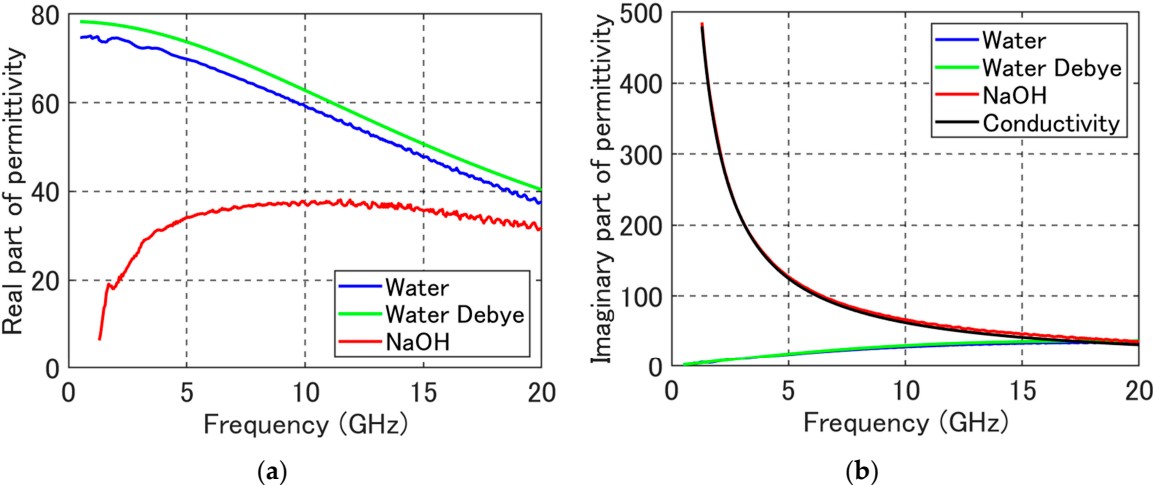

**Figure 5.** Measured results of the (**a**) real and (**b**) imaginary parts of the relative permittivity of the two different liquid samples. The average values of three measurements are plotted. The green lines plot the permittivity of the Debye relaxation model [28]. The black line in (**b**) plots the curve of $\sigma/(2\pi f \varepsilon_0)$ at the conductivity $\sigma$ of 34.7 S/m.

Figure 6 plots the simulated and measured reflection ratios for the applicators using either water or the NaOH solution as samples. Figures 7 and 8 show simulation results for the electric field distribution and the absorbed power distribution, respectively, in the applicator at 2.45 GHz. The absorbed power distribution was obtained by using the calculation function of the specific absorption ratio (SAR) in the HFSS software and was normalized by the material density.

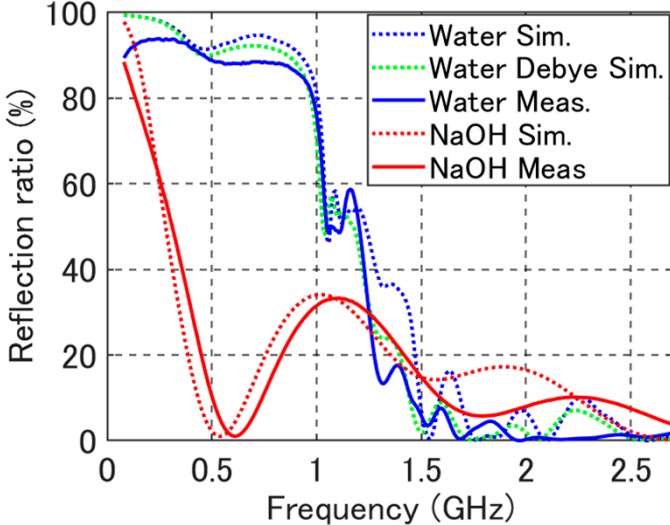

**Figure 6.** Simulated and measured results of the reflection ratios for the developed cylindrical applicator using liquid samples of water and a NaOH solution.

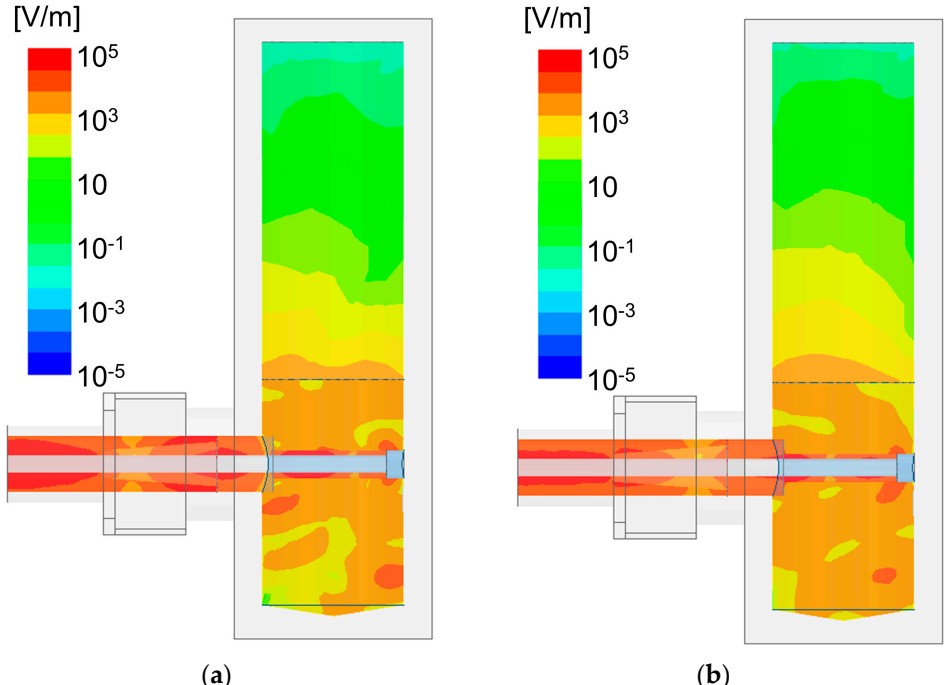

(**a**)  (**b**)

**Figure 7.** *Cont.*

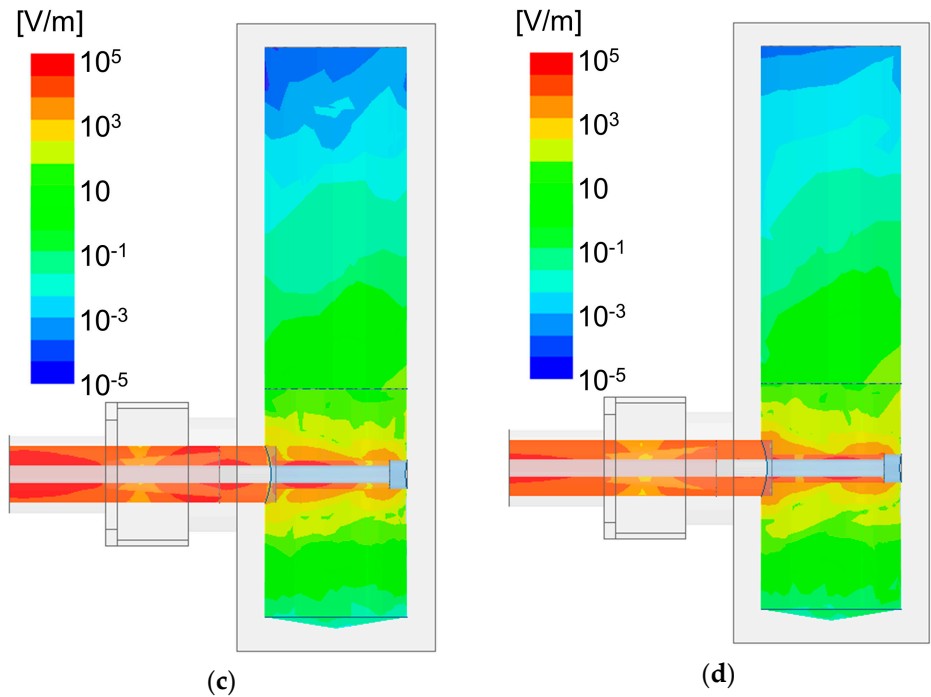

**Figure 7.** Simulated electric field distribution in the cylindrical applicator using liquid samples of water at (**a**) 2.45 GHz and (**b**) 1.7 GHz, and a NaOH solution at (**c**) 2.45 GHz and (**d**) 1.7 GHz.

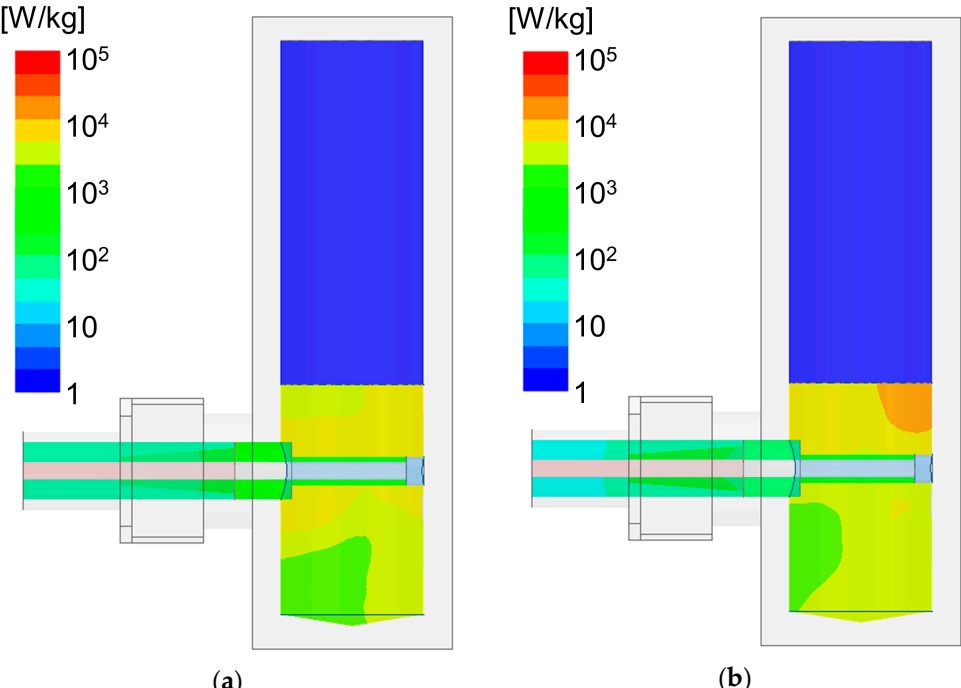

**Figure 8.** *Cont.*

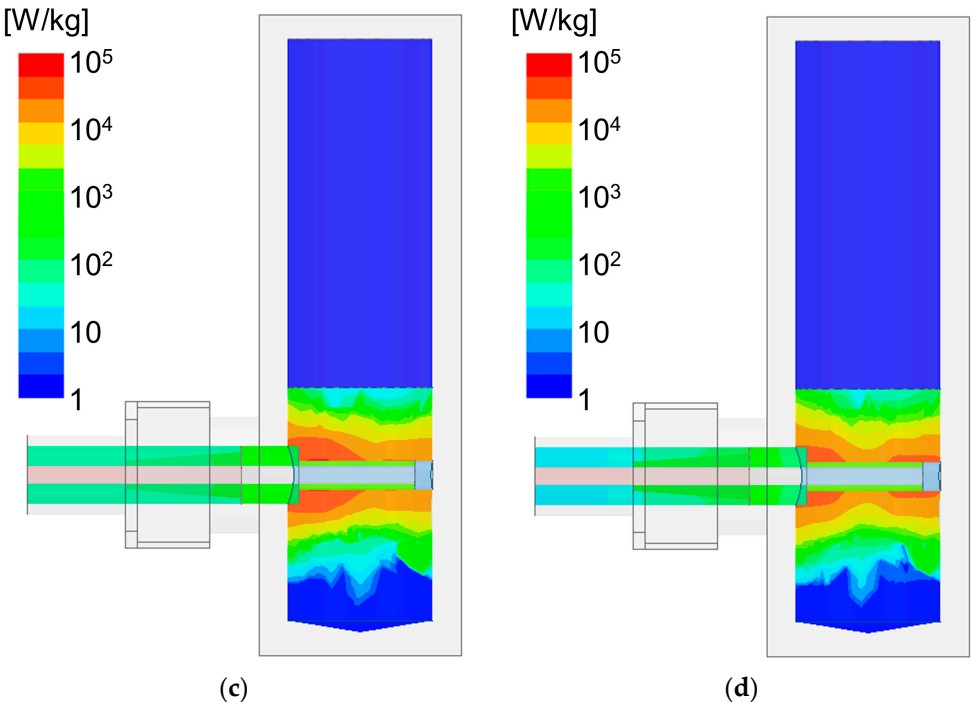

(**c**)                    (**d**)

**Figure 8.** Simulated absorbed power distribution in the cylindrical applicator using liquid samples of water at (**a**) 2.45 GHz and (**b**) 1.7 GHz, and a NaOH solution at (**c**) 2.45 GHz and (**d**) 1.7 GHz.

Figure 9 summarizes the temperature increases observed during the microwave heating trials using water or a NaOH solution. Figure 10 shows the experimental results for the reflection ratio during the microwave heating tests.

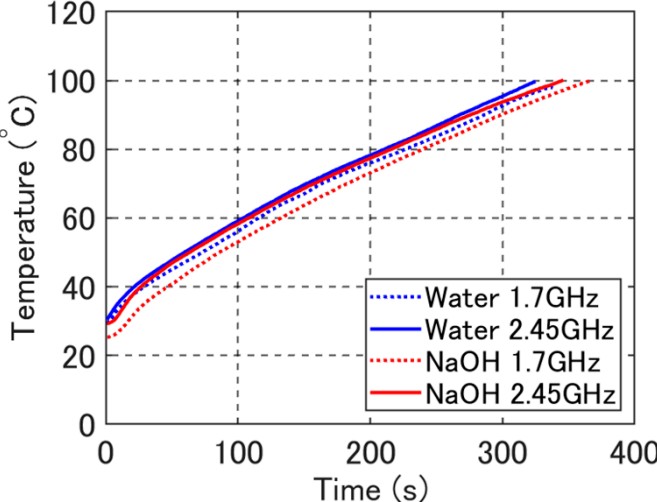

**Figure 9.** Measured results of temperature increases using water and a NaOH solution during microwave heating.

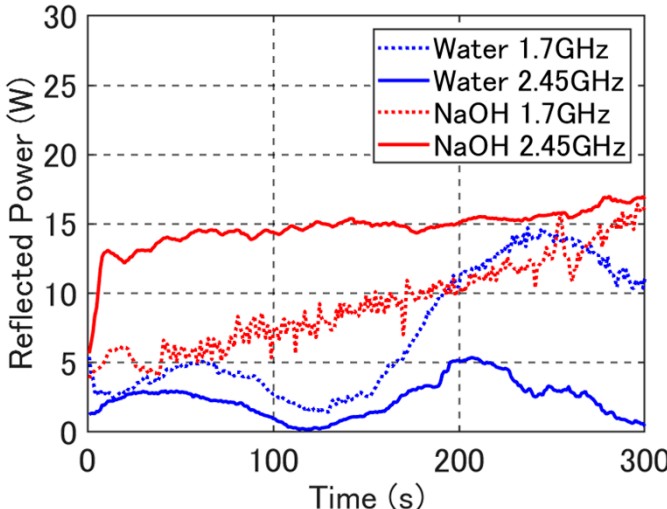

**Figure 10.** Reflected power of the developed cylindrical applicator during microwave heating.

## 4. Discussion

### 4.1. Estimation of the Conductivity and Permittivity

The estimated conductivity of the NaOH solution, which was calculated from Equation (2), is plotted in Figure 11. As it includes the term $2\pi f\varepsilon_0\varepsilon''$ by substituting Equation (1) into Equation (2), the estimated conductivity increases in frequency in the high frequency range. From Figure 11, we adopted the minimum value of the estimated conductivity of 34.7 S/m as the conductivity of the NaOH solution in the simulations, in order to avoid that $\varepsilon''$ becomes negative in the permittivity estimation. The curve of $\sigma/(2\pi f\varepsilon_0)$ at $\sigma = 34.7$ S/m was overplotted as the black line in Figure 5b. It was found, from the comparison between the measured data and the overplotted curve, that the measured imaginary part of relative permittivity for the NaOH solution was almost influenced by the conductivity.

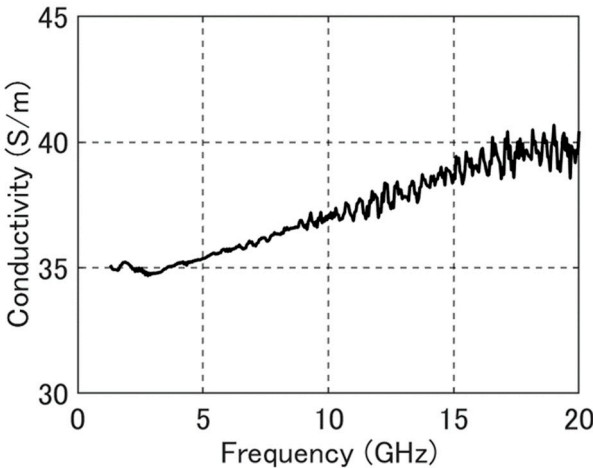

**Figure 11.** Estimated conductivity of the NaOH solution calculated by Equation (2).

Figure 12 shows the estimated imaginary part of the relative permittivity for the NaOH solution, which was calculated from Equation (1). The measured $\varepsilon''$ for the water, whose conductivity was not taken into account, is also plotted. From the measured and estimated results of Figures 5a, 11 and 12, the values of relative permittivity and conductivity taken into the simulation model at 2.45 GHz are summarized in Table 2.

By comparison of the measurement results of water with the Debye relaxation model, the measurement errors of permittivity ranged from −3.7% to −8.4% for the real part, and from 15.2% to −14.5% for the imaginary part when the model parameters described in Section 2.2 were adopted as the true value. As the errors were not negligibly small, the electromagnetic simulations for the water were conducted by using the Debye relaxation model as well as the measured permittivity. The relative permittivity of the Debye relaxation model at 2.45 GHz is also summarized in Table 2.

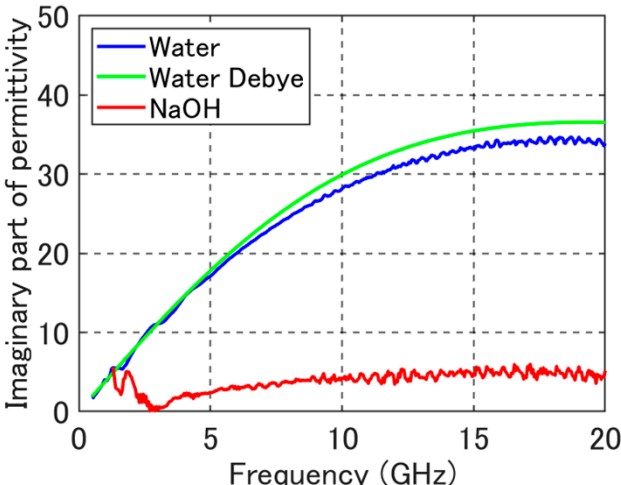

**Figure 12.** Estimated results of the imaginary part of the relative permittivity for the NaOH solution. The conductivity of the NaOH solution was set to 34.7 S/m. The conductivity effect was not taken into account for the water.

**Table 2.** The relative permittivity and conductivity were taken into the simulation model at 2.45 GHz.

| Sample | Relative Permittivity | Conductivity (S/m) |
|---|---|---|
| Water | 74.0–j 9.4 | 0 |
| Water (Debye, 25 °C) [28] | 77.2–j 9.2 | 0 |
| NaOH solution | 22.5–j 2.2 | 34.7 |

### 4.2. Reflection Ratio and Heating Rate of the Developed Applicator

The measurement results for the reflection ratio were in good accordance with the simulation results, as shown in Figure 6. Moreover, the simulation results of the reflection ratios for water when using the Debye relaxation model were in close accordance with those when using the measured permittivity data. The frequency ranges in which the reflection ratio was less than 10% were from 1.45 GHz to 2.7 GHz when using water as the sample and from 1.6 GHz to 2.7 GHz when using the NaOH solution as the sample. The reflection ratio in frequency is dependent on the applicator size and the destructive interference effect inside the liquid sample. Since the diameter of the developed applicator is 25 mm, as shown in Figure 2, the microwaves below 1.45 GHz return to the input port before being propagated and absorbed in the sample.

The electric field and absorbed power distributions in the applicator were relatively uniform in the water case; however, they were focused around the inner conductor in the applicator in the NaOH solution case, as shown in Figures 7 and 8. This indicates that microwaves are well absorbed around the inner conductor due to the conductivity of the NaOH solution. This implies that temperature gradients would take place in the applicator even though the liquid sample was stirred during microwave heating.

Based on the microwave heating results in Figure 9, the heating rate was roughly estimated as 63 to 69 K for a 5 min interval. During the microwave heating trials, the reflected power was less than 17 W, which is equivalent to less than 17% of the reflection ratio, until the time when the

samples reached 100 °C and was gradually increased as the sample temperature increased, as shown in Figure 10. It was reported that the real part of the relative permittivity for the water decreases in temperature [21]. Hence, the reflection ratio increased with heating time as the destructive interference effect inside the liquid sample was enhanced.

The similarity of heating rates shown in Figure 9 was mainly attributed to thermal dissipation from the applicator, in spite of the fact that the reflected power was varied in time and temperature as shown in Figure 10. The developed applicator was not insulated against outer air during the microwave heating tests so that thermal radiation from the applicator occurred at higher temperatures. In addition, thermal conduction occurred from the applicator through the coaxial cable. These thermal dissipation effects would lead to the similarity of temperature increases shown in Figure 9.

Regarding the water, the measurement results of reflected power shown in Figure 10 was compared with simulations at temperatures of 30 °C, 40 °C, 50 °C, and 60 °C. The permittivity of the Debye relaxation model [27] was taken into simulations. The parameters for the Debye relaxation model were obtained from Ref. [28]. The comparison results are summarized in Table 3. The measurement results are comparable with simulations at a temperature of 30 °C, immediately after the start of microwave irradiation; however, the measurement results are smaller than the simulations above 40 °C. This is mainly due to temperature non-uniformity in the applicator. Although the magnetic stirrer was used during microwave heating, temperature non-uniformity would still exist. In addition, numerical simulations are usually much more sensitive than measurements.

**Table 3.** Comparison of reflected power for the water between simulations and measurements.

| Frequency (GHz) | Temperature (°C) | Simulation (W) | Measurement (W) |
|---|---|---|---|
| 1.7 | 30 | 2.2 | 3.9 |
| | 40 | 8.7 | 3.8 |
| | 50 | 16.5 | 4.4 |
| | 60 | 17.9 | 1.9 |
| 2.45 | 30 | 2.2 | 1.3 |
| | 40 | 4.0 | 2.7 |
| | 50 | 5.1 | 2.6 |
| | 60 | 8.6 | 0.7 |

## 5. Conclusions

A microwave irradiation probe for a cylindrical metal applicator was designed and fabricated. The newly developed microwave irradiation probe enables downsizing of the applicator and microwave heating apparatus. The coaxial microwave feeding provides high compatibility with solid-state microwave power amplifiers. Furthermore, the developed microwave irradiation method can be applied to a continuous flow applicator [1] by removing the top and bottom metal shields. The developed applicator will be used for producing high-value added chemicals, for example, production of vanillin from wood particles [30]. Although a magnetic stirrer was used in the developed applicator, the developed applicator should be improved from the viewpoint of temperature uniformity in the liquid sample. Coupled analyses using heat equation and hydrodynamic considerations will provide more precise results, including thermal conduction and thermal dissipation, and will contribute to optimization of the present applicator.

**Author Contributions:** Conceptualization, T.M.; Methodology, T.M., R.N., Y.N., and T.C.; Software, R.N.; Validation, T.M., R.N., N.S., Y.N., T.C., and T.W.; Formal Analysis, T.M., R.N., and N.S.; Investigation, T.M., R.N., Y.N., and T.C.; Resources, Y.N., T.C., and T.W.; Data Curation, T.M., and N.S.; Writing—Original Draft Preparation, T.M.; Writing—Review and Editing, T.M., N.S., and T.W.; Visualization, T.M., and R.N.; Supervision, N.S., and T.W.; Project Administration, T.W.; Funding Acquisition, T.M. and T.W.

**Funding:** This research was funded by CREST Grant Number 1103784, JST, and JSPS Kakenhi Grant Number JP18K04263.

**Acknowledgments:** Measurements of the reflection ratios and the microwave heating tests were conducted through the collaborative research program, Analysis and Development System for Advanced Materials (ADAM), at the Research Institute for Sustainable Humanosphere, Kyoto University.

**Conflicts of Interest:** The authors declare no conflict of interest.

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
