# Peer review of "Development of a Microwave Irradiation Probe for a Cylindrical Applicator"

_processes, doi:10.3390/pr7030143_

Round 1
Reviewer 1 Report
The paper presents a microwave cylindrical applicator with impedance matching through a dielectric structure (ptfe and alumina).
The following points should be addressed:
- Alumina is in direct contact with the material during heating, so it is expected that the heat arrives to the ptfe. Variations in length of the ptfe can be expected. How is this affecting the method? (reflection coefficient, mechanical stability, sealing?)
- How is the magnetic stir bar rotated inside the metal cavity?
- Resonance frequency of the cavity depends on the material. Which range of materials can be heated inside the cavity? The resonance frequency during the whole heating cycle should fall inside the frequency band of the solid state generator.
- The heating of one of the samples is clearly non-uniform. The temperature presented in Fig. 9 could not be representative of the bulk temperature. The temperature at different points of the sample should be provided. If this is not possible, then the sample should be stirred to obtain a homogeneous temperature before taking the temperature measurement. If not, temperature shown in Fig 9 is not real temperature in the whole sample and the heating rate provided by authors is not correct.
Author Response
We would like to express our sincere thanks for your helpful comments. The attached file includes the answers to your comments.

Reviewer 2 Report
Dear editor,
I have read the manuscript entitled “Development of a Microwave Irradiation Probe for a Cylindrical Applicator”. I would recommend the work for publication on the condition that a number of clarifications are included in the manuscript.
In all, the reactor is an innovative concept that can help develop microwave assisted chemistry and heating applications. On a general note though, the intended application, and the specific design choices should be described in more detail to provide context for the work.
Line 43. Other important characteristics of solid state generators besides adjustable frequency are precisely determined and frequency and narrow bandwidth. This should be included.
Line 87–101: Several issues.
1) The Agilent 85070E probe kit can contain any combination of three probes: High Temperature Probe, Slim Form Probe, Performance probe. The manuscript specifies that “a slim dielectric probe” was used. Is this in fact the Slim Form Probe? In order to ensure reproducibility of the work, please use the terminology as defined in the Agilent manual.
2) The default calibration media used with the Agilent 85070E probe kit consist of a shorting block, open probe exposed to air, and water. Since the authors are presumed to using water both as calibration reference and medium, why wouldn’t they directly use reference data for water in your simulation? Or, stated in another way, how can they independently verify whether the measurement method is accurate? A good reference document is the National Physical Laboratory report MAT 23. Another insightful reference is a book by Kremer and Schoenhals entitled “Broadband Dielectric Spectroscopy”. Here additional information on the parametrized fitting models for the dielectric spectra is provided. In addition, I would recommend the authors to please include references to literature data on medium properties.
3) A dielectric spectrum for time harmonic excitations is typically expressed as a summation of complex polynomial fractions. For example a spectrum with two Debye relaxations and one conductivity term could be expressed as (refer to latexbase.com to decode):
$\epsilon(\omega)= \epsilon_{\infty} + \frac{\epsilon_{1}-\epsilon_{\infty}}{1+j\omega\tau_{1}}+ \frac{\epsilon_{2}-\epsilon_{1}}{1+j\omega\tau_{2}} - j\frac{\sigma}{\epsilon_{0}\omega} $
This is not the manner in which the manuscript presents it, so the interrelation of the different phenomena in the spectrum is lost in the polarization behavior of the media.
In addition, since the conductivity in the NaOH solution is mainly due to ionic mobility, Kremer and Schoenhals suggest (eq. 3.18) a modified expression to account for the frequency variation of the conductivity:
$\epsilon(\omega)= \epsilon_{\infty} + \frac{\epsilon_{1}-\epsilon_{\infty}}{1+j\omega\tau_{1}}+ \frac{\epsilon_{2}-\epsilon_{1}}{1+j\omega\tau_{2}} - ja\frac{\sigma}{\epsilon_{0}\omega^{s}} $
I would suggest that the authors try to fit their data to this type of data and tabulate the parameters.
Line 107–114. What references are used to obtain the medium properties data?
Line 134. The stirrer could be included in a separate simulation to verify whether its effect on the overall energy balance and scattering parameters is in fact negligible.
Line 151; Figure 5. For the NaOH solution, the real part of the permittivity seems to approach zero for low frequency. What would be the physical basis for this? Could it be that the dominant effect on scattering is due to conductivity and that this is obfuscating the dielectric polarization in the properties measurement. Could the authors verify in simulation whether the exact value of the real part of the permittivity in this frequency range indeed has little effect on the overall scattering behavior?
Page 6: Figures 6 and 7. The Figure 7 shows what seem to be numerical artifacts around the inner conductor. Has the mesh size been adjusted to see whether these disappear at a finer mesh? Has mesh independence also been verified for the scattering behavior in Figure 6?
Figure 10. Have the authors compared their heating study to simulation? Specifically, the simulation could have been repeated for medium data at higher temperature. This data could have been obtained from literature.
Line 177. There is note of a negative dielectric loss factor (epsilon’’) on this line. What medium model has been used here? This would be an unphysical result.
Line 192–204. The discussion related to the penetration depth is only relevant for a plane wave traveling through a medium without it scattering inside this medium. In the applicator the dimensions are such that scattering is highly likely. In particular, the low rates of dissipation below 1.45 GHz can likely be attributed to a destructive interference effect inside the liquid domain, and not to having too little distance for sufficient dissipation.
Author Response
We would like to express our sincere appreciation for your fruitful comments and suggestions. The attached file includes the answers to your comments.

Reviewer 3 Report
The work done is very interesting and opens many perspectives.
However, there is a lack of information on the conditions of the simulation and in particular on the effect of temperature. I understand that coupling this modeling with the heat equation and hydrodynamic considerations would require a lot of work. However, it is necessary to specify the conditions of the simulation and its limits to offer future readers a more precise result.
The assumptions and details of the simulation are not mentioned. It is impossible to validate the mesh, for example, and to explain the peculiarities of the results obtained, especially the area at the bottom left of the figures 7 (a) and 8 (a).
In the case of NaOH, there are differences of about 10,000 W / kg between the input area of the antenna and the bottom of the cavity. This should result in significant temperature gradients.
If the permittivities depend on the frequency, they also depend on the temperature. At what temperature are the simulations or measurements of Figures 5, 6, 7 and 8?
Figures 8a and b could be changed very significantly by an increase in temperature.
It is therefore very important to clarify all this!
- Was the magnetic stirring bar taken into account in the simulation ?
- What is the effect of the stirrer on the homogenization of the temperature?
- Do you have any idea of the stratification of temperature due to the effects of buoyancy forces ?
- Figures 9 and 10 show results at 1.7 GHz and 2.45 GHz. Figures 7 and 8 show only 2.45 GHz simulation results. It would have been interesting to perform a simulation at the frequency of 1.7 GHz.
- Figure 10 shows differences for reflected powers as a function of frequency. How to explain the similarity of temperature increases in Figure 9 under these conditions? (For mediums as well as for measurements and simulations).
-In the conclusion: it should be emphasized the possible applications in frequency and under variable pressure that constitute an important originality of this device. It should also be noted that this device for the moment is not optimized (with regard to cavity and homogeneity)
Author Response
We would like to express our sincere thanks for your meaningful comments and suggestions. The attached file includes the answers to your comments.

Reviewer 4 Report
The manuscript is on the design of matched coaxial irradiator for heating of water and NaCl solutions. The permittivity of these liquids are frequency- and temperature dependent, and for effective pumping of microwave energy, the matching is needed, at minimum, at room temperature. It is realized using PTFE section, and satisfied matching was reached at 1.6-2.7 GHz. The design is confirmed with numerical simulations and measurements and is interesting in batch mechanically stirred chemistry.
Author Response
We would like to express our sincere appreciation for your comment. We hope the developed applicator will be used for various chemical reactions.
Round 2
Reviewer 3 Report
Thank you for your efforts in these corrections and for clarifying the limits of your study